# Antiseptic Effects and Biosafety of a Controlled-Flow Electrolyzed Acid Solution Involve Electrochemical Properties, Rather than Free Radical Presence

**DOI:** 10.3390/microorganisms10040745

**Published:** 2022-03-30

**Authors:** Alejandro Cabrera-Wrooman, Silvestre Ortega-Peña, Rosa M. Salgado, Belinda Sandoval-Cuevas, Edgar Krötzsch

**Affiliations:** 1Laboratory of Connective Tissue, Centro Nacional de Investigación y Atención de Quemados, Instituto Nacional de Rehabilitación “Luis Guillermo Ibarra Ibarra”, Calzada México-Xochimilco 289, Colonia Arenal de Guadalupe, Tlalpan, Mexico City 14389, Mexico; aca_w@yahoo.com.mx (A.C.-W.); silvestreortega@yahoo.com.mx (S.O.-P.); salgado_rm@yahoo.com.mx (R.M.S.); 2Wound Care Clinic, Hospital General Regional Number 2, Instituto Mexicano del Seguro Social, Calzada de las Bombas 117, Coapa, Girasoles I, Tlalpan, Mexico City 14310, Mexico; cuevasbely@gmail.com

**Keywords:** biofilm, cytotoxicity, oxidation-reduction, wound

## Abstract

Electrolyzed acid solutions produced by different methods have antiseptic properties due to the presence of chlorine and reactive oxygen species. Our aim was to determine whether a controlled-flow electrolyzed acid solution (CFEAS) has the ability to improve wound healing due to its antiseptic and antibiofilm properties. First, we demonstrated in vitro that Gram-negative and Gram-positive bacteria were susceptible to CFEAS, and the effect was partially sustained for 24 h, evidencing antibiofilm activity (*p* < 0.05, CFEAS-treated vs. controls). The partial cytotoxicity of CFEAS was mainly observed in macrophages after 6 h of treatment; meanwhile, fibroblasts resisted short-lived free radicals (*p* < 0.05, CFEAS treated vs. controls), perhaps through redox-regulating mechanisms. In addition, we observed that a single 24 h CFEAS treatment of subacute and chronic human wounds diminished the CFU/g of tissue by ten times (*p* < 0.05, before vs. after) and removed the biofilm that was adhered to the wound, as we observed via histology from transversal sections of biopsies obtained before and after CFEAS treatment. In conclusion, the electrolyzed acid solution, produced by a novel method that involves a controlled flow, preserves the antiseptic and antibiofilm properties observed in other, similar formulas, with the advantage of being safe for eukaryotic cells; meanwhile, the antibiofilm activity is sustained for 24 h, both in vitro and in vivo.

## 1. Introduction

Wound management involves a series of steps focused on wound site preparation and primary or secondary closure, depending on the lesion type. The four fundamental pillars of wound care are the removal of nonviable tissue, management of infection, moisture control, and activation of healthy edges [1]. Lesion repair strategies have been developed based on these principles [2]. Notably, clinicians employ asepsis and antisepsis in the daily practice of wound management. For infection control, the questions of what material to use, the manner of use, and how often to use it are always asked to find the best alternative. Among the antiseptic formulas [3], electrolyzed solutions are increasingly used because they are environmentally friendly sanitizing agents [4]. Having gained wide use in the food industry, these agents are increasingly considered to be safe alternatives for advanced wound management [5]. Several studies have shown that electrolyzed acid solutions have antimicrobial activity against a wide range of Gram-positive and Gram-negative bacteria, fungi, viruses, and even microbial spores [6]. Specifically, electrolyzed acid solutions have been shown to interfere with microbial biofilm formation and to favor its elimination [7], making them desirable for wound management.

Electrolyzed solutions control microbial growth due to the physical and chemical properties acquired during electrolysis. Strong acidic electrolyzed solutions (StAES) produced by different methods and with different materials [4] are characterized by their high oxidation-reduction potential (ORP) and low pH, and their possession of free or combined chlorine ions, hydroxyl free radicals (^•^OH), hydrogen peroxide (H_2_O_2_) [8], and hypochlorous acid; all of these are responsible for the observed antimicrobial activity [9]. The most recognized antiseptic activity of StAES is based on the effect of short-lived free radicals. Despite their transient activity and the microorganism’s own strategies to control the oxidation [10], several authors have shown that their use in early wound management may have advantages over other antiseptic strategies that are more aggressive to tissues and the environment [11]. While a high ORP is a typical characteristic of the StAES, little is known about its long-lasting antiseptic effects, which cause irreversible damage to microorganisms’ vital systems [10]. Despite the usefulness of StAES, it has been reported that mouse fibroblasts [12] and human dental pulp fibroblasts [13] treated with electrolyzed solutions showed similar cytotoxicity to cells exposed to a sodium hypochlorite acid solution, raising doubt about the safety of such solutions for advanced wound management.

In this study, we analyzed the bactericidal effect of an electrolyzed solution produced with a controlled laminar flow method that involves a 2% NaCl solution. The electrolysis equipment features titanium electrodes, covered by platinum or ruthenium and surrounded by a high-density polyethylene membrane that keeps the fluids separate inside the electrolysis chamber. The solution produced by this methodology is called “controlled-flow electrolyzed acid solution” (CFEAS) and belongs to the group of StAES with specific physicochemical properties (ORP > 1100 mV and pH < 3.0) and with a high chloride content (1600 mg/L) [14]. CFEAS has shown utility in the management of wound bed infection at sites of infection recurrence [15]. For that reason, in this work, we studied the ability of CFEAS to eradicate the bacterial biofilm, as well as its cytotoxic effects on human fibroblasts and macrophages. In a pilot trial performed in humans, we treated chronic wounds with CFEAS to understand biofilm control and wound bed preparation.

## 2. Materials and Methods

### 2.1. Bacterial Cultures

The bacteria used in this study were previously reported as biofilm producers [3], and they were identified by an internal code of the Laboratory of Connective Tissue: *Staphylococcus aureus* (B012016), *Enterococcus faecalis* (B022016), *Escherichia coli* (B032016), *Enterobacter cloacae* (B042016), *Klebsiella pneumoniae* (B052016), *Acinetobacter baumannii* (B062016), and *Pseudomonas aeruginosa* (B072016). The different strains were preserved and stored at −70 °C in tryptone soya broth, supplemented with 15% glycerol, until their use for the experiments performed in the present work.

### 2.2. Minimum Inhibitory Concentration Assay

The minimum inhibitory concentration (MIC) was determined using the Clinical and Laboratory Standards Institute protocol [16]. Briefly, each strain was statically cultured in brain–heart infusion broth (BHI) for 24 h at 37 °C. We prepared suspensions of each strain and they were measured in a turbidimeter (HANNA Instruments, Ciudad de Mexico, Mexico) to reach an equivalent of McFarland 0.5 standard (1.5 × 10^8^ colony-forming units [CFU/mL]) in BHI. Every bacterium suspension was 1:20 diluted in BHI broth, and 10 µL of the final suspension, equivalent to 7.5 × 10^4^ CFU, was mixed with 90 μL of Müeller–Hinton broth containing the antiseptics to reach 0.3–95% CFEAS (Electrobioral EJ; Laboratorios Naturales EJ, SA de CV, Guanajuato, Mexico) or 0.3–10% polyhexamethylene biguanide/betaine (PHMB; Prontosan; B. Braun, Melsungen, Germany) as an antiseptic control; cultures in Müeller–Hinton broth served as untreated controls. Experiments were carried out in 96-well, flat-bottomed, sterile plates (Nunc; Thermo Fisher Scientific Inc., Waltham, MA, USA). In order to know that the final inoculum contained the desired CFU, we diluted a further 10 µL of the final suspension in 10 mL of BHI broth, and 100 µL of this suspension was spread over the surface of the BHI agar plate and incubated overnight. Colonies were counted, and only those cultures with the expected CFU were considered for the experiments. MICs were determined by visual readouts of microbial growth. We confirmed our data with a quantitative microbial growth analysis by inoculating 10 μL of the bacterial culture medium from MIC assay in plates with 5% bovine blood agar (BD Diagnostics, Sparks Glencoe, MD, USA) and further incubated at 37 °C for 24 h. The absence of CFU was confirmatory of visual readouts. We conducted two separate experiments with five replicates each. All the bacteriological experiments were carried out under sterile conditions, including a sterile hood, and culture media.

### 2.3. In Vitro Eradication of Single-Species Biofilms

For biofilm eradication, every bacterium culture was grown at 37 °C for 24 h in BHI broth supplemented with 2% glucose (BHI-G). Then, 7.5 × 10^4^ CFU in 100 μL of BHI-G was distributed into each well of a 96-well, flat-bottomed sterile plate, and they were statically incubated at 37 °C for 24 h. Once the biofilm had formed, the medium was gently removed, and the wells were washed three times with saline to remove free bacteria. Then, 100 μL of 95% CFEAS or 1.5% PHMB in BHI-G was added and plates were statically incubated at 37 °C for 3, 6, 12, and 24 h. Chosen concentrations for CFEAS and PHMB were derived from MIC assays, as well as the previously observed effects for PHMB [3]. The culture medium was decanted, and each well was washed three times with saline solution and allowed to dry at room temperature for 2 h. Residual biomasses were stained on every well for 20 min with 200 μL of 0.1% crystal violet staining solution. The dye was then eliminated, and each well was washed three times with saline solution. Finally, the purple dye was extracted with 180 μL of 95% ethanol for 30 min, and the optical density (OD) was quantified at 540 nm on an ELISA microplate reader (xMark Bio-Rad Laboratories, Inc., Hercules, CA, USA). The OD value obtained for culture in the absence of an antiseptic (untreated control) was considered 100% of the biomass. Values below the control behavior represent biofilm eradication [3]. We conducted two separate experiments with three replicates each. All bacteriological experiments were carried out under sterile conditions, including a sterile hood, and culture media.

### 2.4. Cell Cultures

The BJ human fibroblast and THP-1 macrophage cell lines were kindly donated by Professor Ivan Velasco from the Instituto de Fisiología Celular, UNAM, Mexico and Professor Enrique Ortega Soto from the Instituto de Investigaciones Biomédicas, UNAM, Mexico, respectively. Fibroblast cultures were grown in Dulbecco’s modified Eagle medium (Gibco; Life Technologies, Grand Island, NY, USA) supplemented with 10% fetal bovine serum (Gibco), 2 mM L-glutamine (Gibco), 100 U/mL penicillin, and 100 μg/mL streptomycin (Gibco). Macrophage cultures were grown in RPMI-1640 (Gibco) supplemented with 10% fetal bovine heat-inactivated serum, 1 mM sodium pyruvate (Gibco), 0.1 mM non-essential amino acids (Gibco), 0.1 mM L-glutamine (Gibco), 100 U/mL penicillin, and 100 μg/mL streptomycin (Gibco). Cells were grown in an incubator at 37 °C and 5% CO_2_.

### 2.5. Cytotoxicity Assessment on Eukaryotic Cells

Twenty-five thousand BJ fibroblasts or THP-1 macrophages per 200 μL were plated in every well from a 96-well plate (Nunc) with their corresponding culture medium. Plates were incubated at 37 °C and 5% CO_2_ for 24 h. Then, media were replaced with 200 μL of the corresponding culture media with the different concentrations of CFEAS (1–95% *v*/*v*). After 6 or 24 h incubation at 37 °C and 5% CO_2_, 10 μL of a 5-μg/mL solution of 3-(4,5-dimethylthiazol-2-yl)-2,5-diphenyltetrazolium bromide (MTT Invitrogen Corp., Carlsbad CA, USA) was added, and the plates were incubated for an additional 3 h. The culture medium was removed, and each well was washed three times with a phosphate-buffered saline (PBS) solution. A reduction in MTT was evident after the dissolution of the formazan crystals with 200 μL of dimethyl sulfoxide/isopropanol and was measured by colorimetry at 570 nm with an iMARK Microplate Absorbance Reader (Bio-Rad Laboratories, Inc.). We conducted two separate experiments with three replicates each.

### 2.6. Superoxide Ion Activity Assessment

Twenty-five thousand BJ fibroblasts or THP-1 macrophages per milliliter were plated in every well from a four-well cell-culture chamber slide with their corresponding culture medium (Falcon; Becton Dickinson Labware, NJ, USA) and incubated the cultures at 37 °C and 5% CO_2_ for 24 h. Media were replaced with 1 mL of the corresponding culture media with different CFEAS concentrations (1%, 5%, 10%, and 50% *v*/*v*); media without CFEAS were included for control wells. Chamber slides were incubated at 37 °C and 5% CO_2_ for 6 and 24 h. The ninety-five percent CFEAS concentration was excluded from the experiment due to its lethal activity on eukaryotic cells. After these periods, we incubated at 37 °C and 5% CO_2_ for an additional 4 h period, after which media were replaced by fresh media with different concentrations of CFEAS for the experimental group: The culture media with 500 µM H_2_O_2_ for positive controls of oxidation and media without treatment for untreated controls. The cultures were washed twice with PBS, treated with 1 mL 5-μM 5-(6′-triphenylphosphoniumhexyl)-5,6-dihydro-6-phenyl-3,8-diamino phenanthridine (MitoSOX Red stain; Invitrogen) in PBS, and incubated for another 15 min. Cell monolayers were washed twice with PBS and fixed with 4% paraformaldehyde for further cytofluorescence analysis. After two washes, they were then mounted with VECTASHIELD mounting medium with DAPI (Vector Laboratories, Inc., Burlingame, CA, USA). The MitoSOX staining showed orange fluorescence after oxidation with ^•^O_2_^−^ (excitation 510 nm, emission 580 nm), and DAPI enabled the identification of cell nuclei in blue (excitation 360 nm, emission 460 nm). Images were recorded using a microscope (Axio Imager.Z1; Carl Zeiss, Göttingen, Germany) fitted with a monochromatic high-speed camera (Axiocam; Carl Zeiss). Fluorescence densitometric evaluation was performed via digital image analysis. We conducted two separate experiments, and we assessed five random cells in equivalent central fields for every culture using the AxioVision digital image-processing software (ver. 4.8.1.0; Carl Zeiss) [17].

### 2.7. Human-Infected Subacute and Chronic Wounds Treated with CFEAS: A Pilot Trial

In order to demonstrate the in vivo reduction in total microbiota and biofilm presence after CFEAS treatment, we performed an evaluation of total CFU/g of tissue, as well as a histological architecture of the biofilm in infected subacute and chronic wounds from five patients. The criteria we followed to consider a chronic wound were based on those wounds that did not progress through a normal sequence of repair after 3 months and/or the wounds that lacked a 40% reduction after 4 weeks of optimal treatment. All the participants completed an institutional consent form for routine ambulatory procedures according to the Clinical Practice Guidelines of the Wound Care Clinic at Hospital General Regional Number 2, Instituto Mexicano del Seguro Social, Mexico City. The samples obtained for this evaluation came from patients with venous leg ulcers or soft tissue from subdermal damage. Patients were not under any specific treatment, locally or systemic, for at least one and a half months prior to the study, except for analgesics in the case of pain (paracetamol or naproxen). Early wound management included local anesthesia and surgical cleaning with saline to remove biofilm excess and necrotic tissue. A three-millimeter-diameter, full-thickness punch biopsy was obtained to routinely assess total CFU; samples were handled in Stuart’s medium for transportation and processed immediately.

After initial sampling, a gauze wet with enough 100% CFEAS to thoroughly impregnate the wound was placed and covered with Tegaderm transparent film dressing (3M, Minneapolis, MN, USA); a bandage was placed over the first dressing to protect it; only one CFEAS treatment was administered during the 24-h period. Wounds were undressed and gently rinsed with saline. A new biopsy was obtained, and tissue was handled as before. For both samples, and before grinding the tissue for CFU quantitation, a transversal section of every biopsy was separated and fixed in Protocol SAFEFIX II (Fisher Diagnostics, Portage, MI, USA). Fixed tissues were embedded in paraffin, and 5-μm transversal sections were stained with hematoxylin and eosin to evaluate biofilm residues and cellular infiltrate using an Axio Observer Z1 microscope (Carl Zeiss), fitted with a high-speed polychromatic camera (AxioCam; Carl Zeiss).

### 2.8. Determination of the Bacterial Colony Forming Units per Gram of Tissue

Under sterile conditions, tissues were weighed and then ground with 5 mL of 0.85% saline solution. Subsequently, aliquots of 200 μL, 50 μL, and 10 μL were plated in trypticase soy agar plates and cultured for 48 h at 37 °C. After incubation, the microbial colonies were counted, with the results recorded as CFU/g of tissue. Logarithm values were plotted, and before and after treatment groups were compared. All bacteriological experiments were carried out under sterile conditions, including a sterile hood, and culture media.

### 2.9. Statistical Analysis

Results were analyzed using GraphPad InStat 3.0 (GraphPad Software, Inc., San Diego, CA, USA). General differences in the eradication of biofilm assay were evaluated with a nonparametric analysis of variance (Kruskal–Wallis Test), and differences among bacterial cultures were evaluated using Dunn’s or Tukey–Kramer multiple-comparisons tests, according to the normality behavior of the data. For cytotoxicity and ^•^O_2_^−^ activity, we performed one-way ANOVA analysis, and differences among cultures were evaluated using Bonferroni’s multiple comparison test. A comparison was made of clinical data before and after CFU/g of tissue, using a two-tailed paired *t*-test, since samples were paired, and all values exhibited Gaussian distributions. *p*-values ≤ 0.05 were considered to be significant.

## 3. Results

### 3.1. CFEAS Exhibits Bactericidal and Antibiofilm Effects

First, we evaluated the antiseptic properties of CFEAS on different planktonic bacteria derived from clinical isolates (Appendix A). We observed that Gram-negative and Gram-positive bacteria were susceptible to the treatment with CFEAS at concentrations higher than 95%. The effect of CFEAS contrasted with PHMB (antiseptic control), which was effective from concentrations of 1.5% (Table 1).

At 3–6 h, the CFEAS had a statistically significant effect on the partial eradication of mature biofilm from Gram-negative bacteria, particularly *K. pneumoniae* when compared to the control. The effect of the CFEAS changed over time, with late anti-biofilm activity in some biomasses formed by *S. aureus* and *E. cloacae* and a transitory effect for the other organisms (Figure 1). At 24 h, the effect of the CFEAS was very similar to that observed for 1.5% PHMB.

### 3.2. Fibroblasts Are Less Sensitive to CFEAS Induced Cytotoxicity than Macrophages

In fibroblast cultures, the tested CFEAS concentrations (1–95%) exhibited a cytotoxic effect after 6 h of treatment, with 50% CFEAS (*p* ≤ 0.05). Although the average relative cell viability was diminished after 24 h of fibroblast culture in the presence of 50% CFEAS, we did not find a significant cytotoxic effect (Figure 2). Macrophage cultures treated with all CFEAS concentrations, except 10%, showed reductions in cell viability after 6 h treatment (*p* ≤ 0.05. Figure 2) and the effect disappeared at 24 h. Thus, the CFEAS may only have a cytotoxic effect in the early stage, which may be associated with its contribution to oxidation conditions. A peculiarity observed in macrophage behavior was a rise in formazan formation when cells were treated with 10% CFEAS, although there was no statistically significant difference from the control. However, we performed a proliferation study based on bromodeoxyuridine (BrdU) uptake in all the cultures and did not observe any change in cell proliferation among the different groups (Appendix A).

### 3.3. Fibroblasts Compensate More Efficiently CFEAS Derived Oxidizing Radicals than Macrophages

Fibroblasts were more resistant to the oxidizing effects derived from CFEAS treatment than macrophages. Fibroblasts treated with different concentrations of CFEAS exhibited ^•^O_2_^−^ activity levels that were below those of untreated cultures after 6 h of treatment in a CFEAS-concentration-dependent fashion (*p* < 0.05; treated vs. untreated, and *p* < 0.05; 1% vs. 5 and 10% CFEAS). Meanwhile, macrophages showed the effect only at 1% CFEAS at the same timepoint. Otherwise, after 24 h treatment, only the 50% CFEAS promoted ^•^O_2_^−^ activity in both cell cultures (*p* ≤ 0.05. Figure 3 and Figure 4). The treatment of fibroblasts with H_2_O_2_ (positive control of oxidation) significantly increased ^•^O_2_^−^ activity after 6 h; this effect disappeared at 24 h. The reverse effect was observed for macrophages (Figure 3 and Figure 4).

### 3.4. Treatment with CFEAS of Subacute and Chronic-Infected Wounds Reduced Significantly Biofilm Formation

Subacute and chronic wounds from five adult patients with an average evolution time of 6.8 years (range 18 days–16 years) were cleansed and then treated for 24 h with gauze impregnated with 100% CFEAS. Clinically, patients showed changes in wound bed characteristics after treatment (Appendix A). First, it was evident that biofilms removed during the first washing were not reorganized in treated wounds, since the wound bed and borders were free of the previously observed bioburden. Granulation tissue looked like a homogeneous reddish material, and, interestingly, wound exudate was controlled, although injuries remained moist (Figure 5a,b). Histological sections of wounds treated for 24 h showed a decrease in biofilm thickness. Before treatment, biofilm structure was observed as a superficial eosinophilic-thickened matrix (Figure 5c, frame), with embedded monocytes and neutrophils (Figure 5c); meanwhile, after CFEAS treatment, the microbial matrix was thin and sparse, with isolated leukocytes (Figure 5d). Colony-forming units showed a statistically significant reduction after 24-h CFEAS treatment (* *p* ≤ 0.05. Figure 5e), correlating with histological and clinical images.

## 4. Discussion

The use of electrolyzed solutions for the treatment of infected wounds was first considered more than 20 years ago [18]. These solutions appear to have antiseptic effects against a wide variety of microorganisms [19] and leave no toxic residues that might compromise healing [18]. Although multiple studies have shown that electrolyzed solutions provide a therapeutic alternative for wound management [15], some reports are contradictory, and the analyses of experimental cytotoxicity data for electrolyzed solutions are not sufficiently supported by statistical results [20]. In this study, we found that CFEAS, an StAES, had deleterious effects on Gram-positive and Gram-negative bacteria in a planktonic state and biofilms. We found a greater antiseptic effect on Gram-negative bacteria that were starting to form biofilms, which increased over 24-h treatment, while the effect was also evident in Gram-positive bacteria. Studies of mature biofilm removal with several antiseptics have regularly shown that the microbiocidal effect is limited to the early stages of treatment and that the microbial biomass at least partially recovers as the antiseptic effect declines over time [3]. A limitation of CFEAS was that it only demonstrated effectiveness in concentrations higher than 95%; this means that CFEAS is a ready-to-use solution, whereas other electrolyzed solutions, such as Envirolyte^®^ electrolyzed water, can work at concentrations that are 10 times lower. However, comparisons should also consider that CFEAS presents 500 times less residual free chlorine than Envirolyte^®^ [19], and that the demonstrated effect for the latter is only proven for up to 30 min after treatment in vitro, suggesting that the main antiseptic effect of Envirolyte^®^ and others is due to short-lived free radicals derived from HClO. Our data showed that a single dose of CFEAS sustained an equal or greater effect for 24 h, which makes it more practical for wound management cases in which the application of treatment several times a day is less feasible. These data can be supported by other microbiocidal properties derived from the physicochemical characteristics of the solution, such as the low pH and high ORP [10]. The debridement of wounds with biofilms is imperative; however, the rebuilding of biofilms is frequently observed when the patient is not subsequently treated with antibiofilm therapy [21]. The CFEAS showed moderate antiseptic effects on mature biofilms, similar to other StAES reported by different authors [7]. However, the significant reduction in the biomass observed in biofilms treated in vitro and in vivo with CFEAS suggests that the synergistic effect of wound debridement followed by CFEAS treatment could avoid biofilms being rebuilt.

Generally, antisepsis derived from StAES is related to the toxicity of the short-lived free radicals (e.g., Cl^−^, ^•^OH, ^•^O_2_^−^) present in the solution, which might have a time-limited effect, or their microbiocidal properties being destroyed by exudate-containing molecules [8]. Although H_2_O_2_ is a source of ^•^OH in StAES, its bactericidal activity contributes less than HClO and molecular chlorine (Cl_2_) [8]. Therefore, we must recognize that much of the antimicrobial (antiseptic) effect of electrolyzed solutions is due to the oxidation-reduction conditions of their production [22,23]. In addition, the CFEAS presents very low levels of residual-free chlorine, which might be due to the time and temperature used for electrolysis [14,24], so CFEAS’s sustained antiseptic activity could come from another physicochemical property derived from the electrolysis procedure, such as the ORP. These results are supported by other reports showing that the ORP of the medium is critical for microbial growth. The variety of acidity indicators and dyes used as antiseptics since early in the last century provide examples of this; their mechanism of action is simply the generation of redox states that are deleterious to microorganisms, ranging from bacteriostatic to antiseptic [25].

It is known that healing is promoted under reductive conditions [26], and high levels of mitochondrial reactive oxygen species (ROS) are considered to be very important for the induction of apoptosis [27]. Thus, electrolyzed acid solutions that work under oxidative conditions would definitely be cytotoxic to the cells involved in repair processes [9], perhaps beyond the toxicity caused by chemically short-lived antiseptic substances [3]. Although it has been reported that the decrease in intracellular ROS by electrolyzed water might be due to the activation of intracellular antioxidant systems via indirect ROS scavenging mechanisms by the cells [28], these data only refer to alkaline electrolyzed solutions or electrochemically reduced waters [29]. However, we observed that, mainly in fibroblasts, ^•^O_2_^−^ activity diminished after 6 h of treatment with CFEAS, suggesting a cell compensatory mechanism of antioxidant molecules (e.g., superoxide dismutase) triggered by the ^•^O_2_^−^ presence [30]. This finding is very important because another source of ROS can be derived from the electrolysis process in the StAES or by the persistent intracellular ^•^O_2_^−^ generation induced by the H_2_O_2_ contained in them [31]. Additionally, we observed that fibroblasts treated with CFEAS were less sensitive than macrophages, indicating that the antiseptic treatment leads to limited damage to these dermal resident cells during wound repair. In such a case, our results suggest that the use of CFEAS is a safe strategy for the management of infected wounds, as it has limited cytotoxic effects in fibroblasts and macrophages.

These findings suggest the presence of mechanisms regulating redox states in eukaryotic cells with greater efficiency than that observed in prokaryotic cells [32], or eukaryotic resistance associated with reductive mechanisms at the cytoplasmic, nuclear, and, particularly, mitochondrial levels [33], despite the conditions created by the CFEAS.

Other authors have argued that some ROS present in electrochemically produced anode solutions might trigger early healing via fibroblast migration and proliferation [11]. We could not demonstrate such an effect in our in vitro model, but this might be the case for in vivo systems, in which ROS are known to contribute to the epithelial response via prostaglandin E2 induction [34].

Clinically, CFEAS has been assayed in healthy hallucal human skin, where it has demonstrated bacteriostatic and, in some cases, bactericidal effects [35]. Those data have the limitation that the test was performed on healthy skin, where microbial growth is, per se, limited by the own local microbiota; additionally, normal skin does not allow for biofilm formation in healthy conditions. However, in this work, we demonstrated a bactericidal effect and biofilm control by CFEAS treatment of human subacute and chronic wounds, indicating that, even in the worst-case scenario, CFEAS can control infection and biofilm formation, as was reviewed recently by Yan P et al. [36].

## 5. Conclusions

We demonstrate that CFEAS, an StAES prepared by laminar flow during electrolysis, has limited cytotoxic effects in eukaryotic cells, perhaps through a self-compensatory mechanism, but has the ability to control microbial and biofilm growth, which might be of great benefit during wound-healing. With this work, we suggest that the combination of oxidant reactive species (early) and electrochemical properties (all the time) of the CFEAS means that the solution exhibits an antiseptic synergistic effect, which is useful for cleansing wounds and burns. An investigation of the underlying mechanism of CFEAS in preclinical models, as well as in randomized, controlled, prospective, and comparative trials in humans, are required to understand how antisepsis and redox state control work together during wound healing.

## Figures and Tables

**Figure 1 microorganisms-10-00745-f001:**
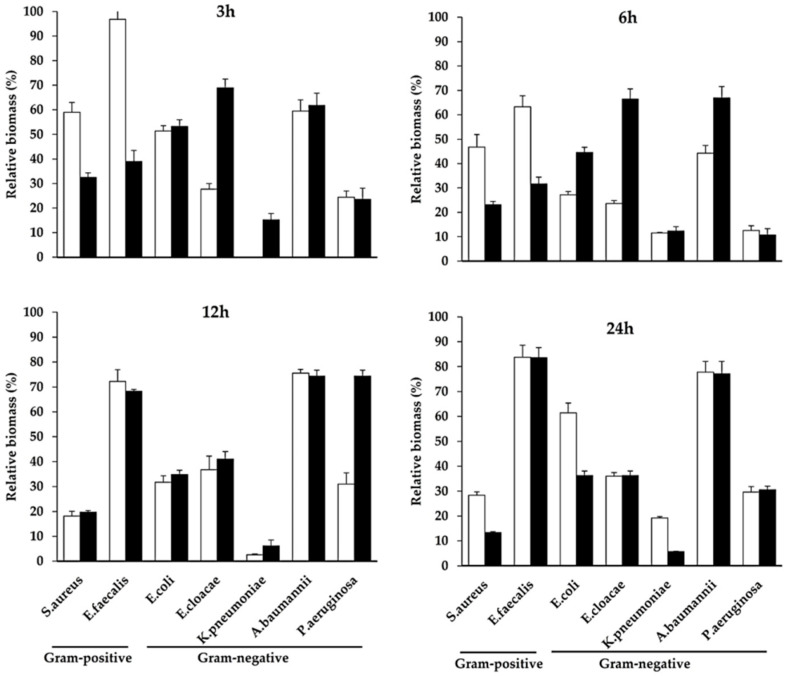
In vitro evaluation of mature biofilm eradication. Remaining biomass on the culture plates was evaluated after 3 h, 6 h, 12 h, and 24 h of in vitro treatment. Values represent the percentage of the mean ± SD of biomass formation, relative to untreated control, in cultures treated with 95% CFEAS (white bars) or 1.5% PHMB (black bars, antiseptic control). Differences in the eradication of biofilm were evaluated with a nonparametric analysis of variance (Kruskal–Wallis Test), and differences among microbial cultures were evaluated using Dunn’s or Tukey–Kramer multiple-comparisons tests, according to the normality behavior of the data. All treatments exhibited statistically significant differences compared to the corresponding untreated control (*p* ≤ 0.05), except for *E. faecalis* CFEAS treated cultures at 3 h.

**Figure 2 microorganisms-10-00745-f002:**
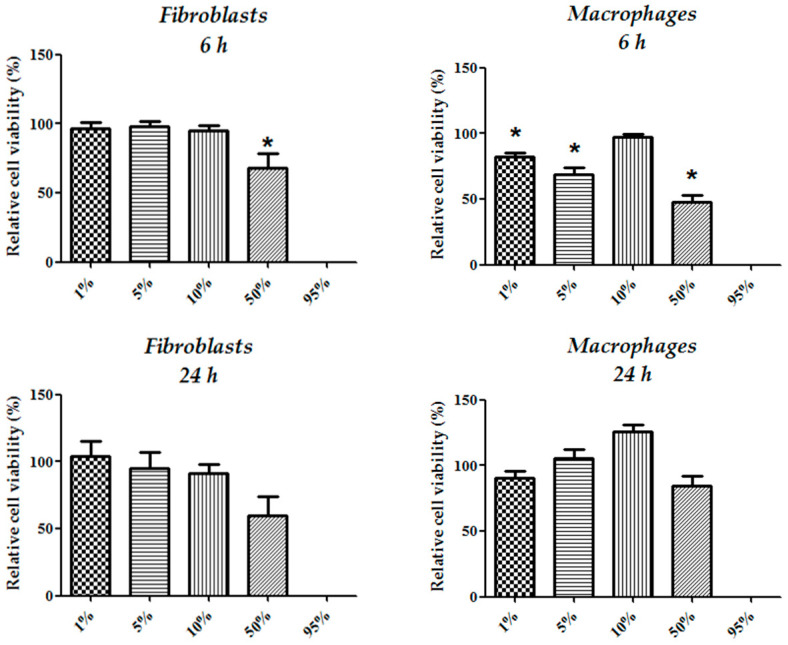
Assay of cell viability in fibroblast and macrophage cultures. MTT reduction was evaluated in fibroblast and macrophage cultures at 6 h and 24 h, respectively. Values represent the percentage of cell viability of cultures treated with CFEAS at different concentrations (1–95%) from the untreated control (mean ± SD). The treatment of fibroblasts and macrophages with 50 and 95% or 1, 5, 50, and 95% CFEAS, respectively, showed statistically significant differences when compared with untreated controls (* *p* ≤ 0.05).

**Figure 3 microorganisms-10-00745-f003:**
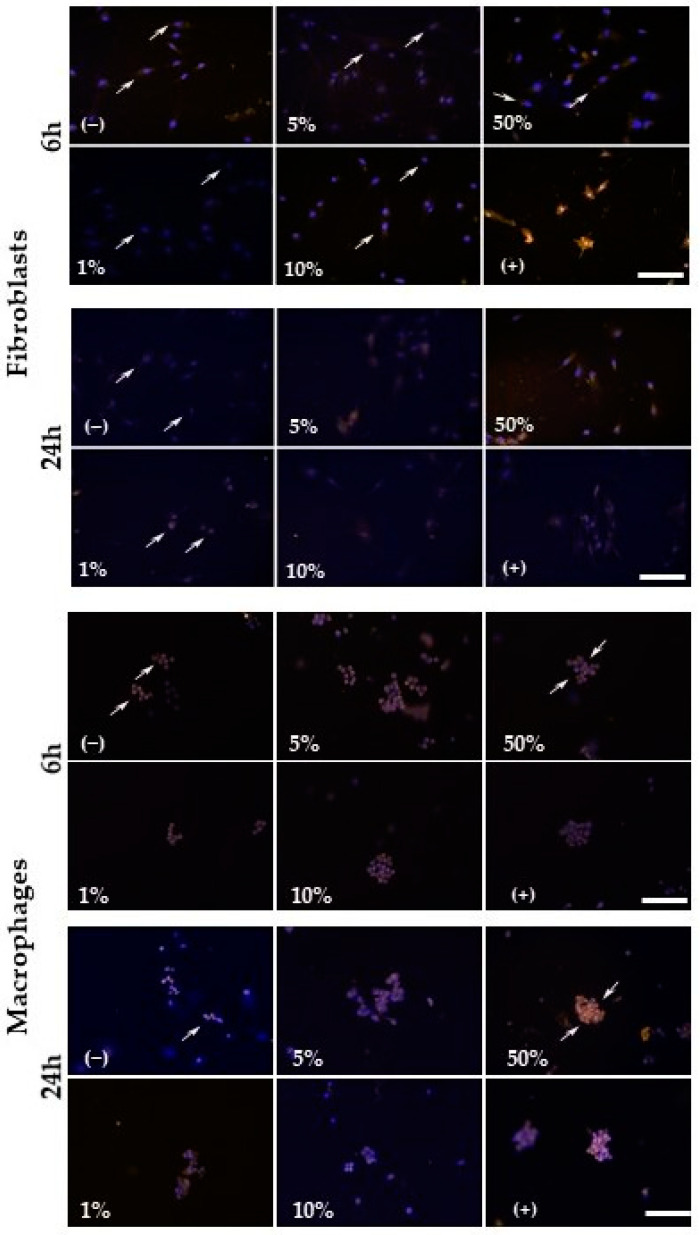
Fluorescence photomicrographs from fibroblast and macrophage cultures. ^•^O_2_^−^ activity was evidenced by MitoSOX Red staining at 6 h and 24 h. Untreated cultures, cultures treated with different concentrations of CFEAS (1%, 5%, 10%, and 50%), and cultures treated with 500 μM H_2_O_2_ (positive control of oxidation). Arrows indicate ^•^O_2_^−^ activity (orange), cell nuclei in blue. Bar = 50 μm.

**Figure 4 microorganisms-10-00745-f004:**
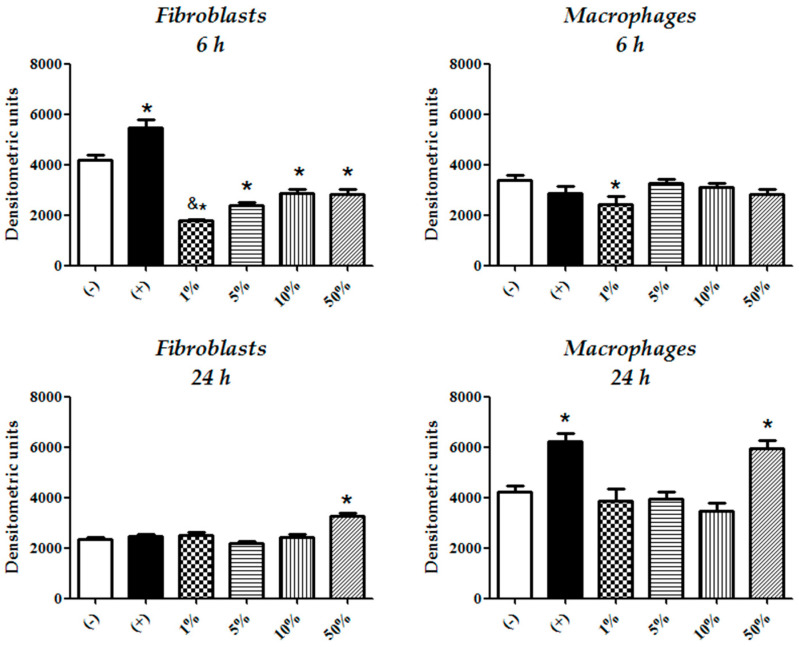
Assays of ^•^O_2_^−^ activity in fibroblast and macrophage cultures. MitoSOX fluorescence evaluation was performed by image densitometric analysis (mean ± SD) from fibroblast and macrophage cultures after 6 h and 24 h of treatment with CFEAS. Results are shown for untreated cells (−; base cell oxidation), cells treated only with H_2_O_2_ (+; positive control of oxidation due to the induction of the ^•^O_2_^−^ activity), and cells treated with 1–50% CFEAS. Statistically significant differences were observed between untreated cells (−) and fibroblast cultures treated for 6 h with 1–50% CFEAS (* *p* ≤ 0.05, control vs. 1, 5, 10, and 50%). One percent vs. 10 and 50% CFEAS also exhibited statistically significant differences (*p* ≤ 0.05) in fibroblast cultures. After 24 h of fibroblast treatment, only 50% CFEAS was statistically significantly different from the control (−) (* *p* ≤ 0.05). Fibroblasts treated for 6 h with H_2_O_2_ (+) exhibited a statistically significant difference when compared to untreated control (−) (* *p* ≤ 0.05). For macrophage response, after 6 and 24 h of treatment with 1% or 50% of CFEAS, respectively, statistically significant differences were seen when compared with untreated cells (−) (* *p* ≤ 0.05). Macrophages treated for 24 h with H_2_O_2_ (+) exhibited a statistically significant difference when compared to untreated control (−) (* *p* ≤ 0.05). Differences in ^•^O_2_^−^ activity were assessed by one-way ANOVA analysis, and differences among cultures were evaluated using Bonferroni’s multiple-comparison test. * *p* or ^&^
*p*-values ≤ 0.05 were considered to be statistically significant.

**Figure 5 microorganisms-10-00745-f005:**
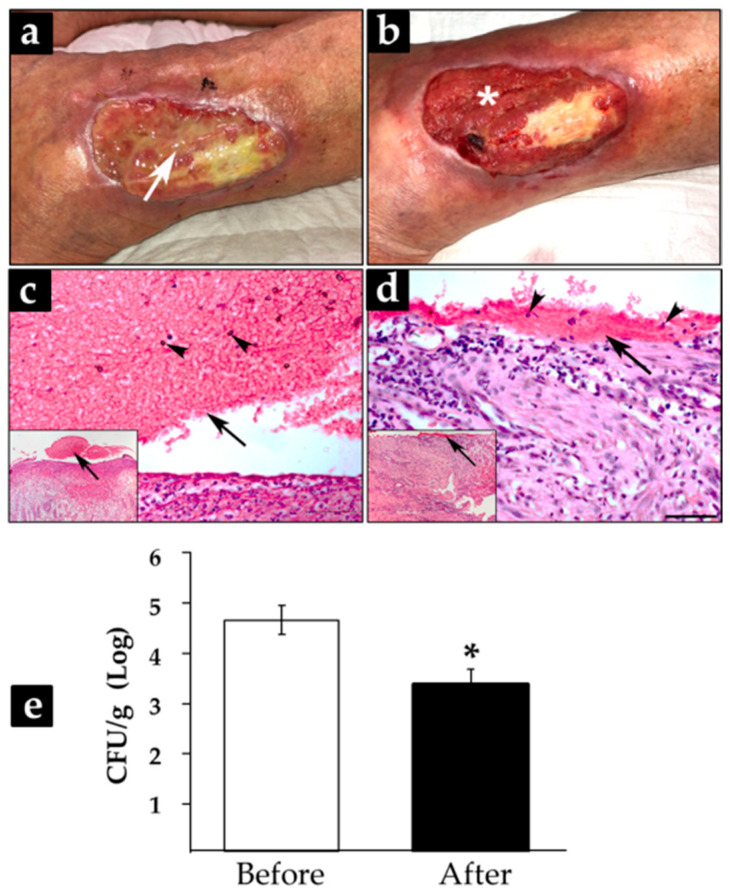
Effects of treatment with CFEAS on human infected wounds. Clinical pictures from a 73-year-old representative patient with a venous leg ulcer with 16 years of evolution time. Wound before treatment (**a**) and after 24 h of CFEAS treatment (**b**). Biopsies were obtained from wounds and stained by hematoxylin and eosin techniques; (**c**,**d**) representative sections before and after treatment, respectively; bar = 100 µm. White arrow in (**a**) shows biofilm and white asterisk in (**b**) shows the granulation tissue. Black arrows show biofilm and black arrowheads inflammatory cells (**c**,**d**). With the aim of providing a panoramic view of the sections, photomicrographs with low magnification were included in a frame. Colony-forming units were assessed from the biopsies of the five patients before and after treatment, and they were plotted in (**e**); differences in log CFU/g of tissue were analyzed by two-tailed paired *t*-test; * *p*-value = 0.0004.

**Table 1 microorganisms-10-00745-t001:** Analysis of the minimum inhibitory concentration of bacterial growth in planktonic conditions.

Strain	CFEAS (%)									PHMB (%)		
	95	50	25	12.5	6.2	3.2	1.5	0.7	0.3	1.5	0.7	0.3
**Gram-positive**												
*S. aureus*	**−**	**+**	**+**	**+**	**+**	**+**	**+**	**+**	**+**	**−**	**−**	**−**
*E. faecalis*	**−**	**+**	**+**	**+**	**+**	**+**	**+**	**+**	**+**	**−**	**−**	**−**
**Gram-negative**												
*E. coli*	**−**	**+**	**+**	**+**	**+**	**+**	**+**	**+**	**+**	**−**	**−**	**−**
*K. pneumoniae*	**−**	**+**	**+**	**+**	**+**	**+**	**+**	**+**	**+**	**−**	**−**	**−**
*E. cloacae*	**−**	**+**	**+**	**+**	**+**	**+**	**+**	**+**	**+**	**−**	**−**	**−**
*A. baumannii*	**−**	**+**	**+**	**+**	**+**	**+**	**+**	**+**	**+**	**−**	**−**	**−**
*P. aeruginosa*	**−**	**+**	**+**	**+**	**+**	**+**	**+**	**+**	**+**	**−**	**+**	**+**

+ growth of bacteria, − no growth of bacteria.

## Data Availability

Not applicable.

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
