# Peer review of "Antiseptic Effects and Biosafety of a Controlled-Flow Electrolyzed Acid Solution Involve Electrochemical Properties, Rather than Free Radical Presence"

_microorganisms, 2022, doi:10.3390/microorganisms10040745_

Round 1

Reviewer 1 Report

The aim of the study was to investigate the antiseptic activity of a controlled-flow electrolyzed acid solution and biosafety of CFEAS which rather present electrochemical action than free radical presence. The study was performed on Gram-positive and Gram-negative bacteria, at different stages of biofilm formation as well as on cell lines and in vivo wounds. I think a lot of effort went into the study. However, additional experiments and controls (conducted also before the actual in vitro/in vivo study) are needed to give a fool view of the matter. Unfortunately, in my opinion the research was not conducted correctly. Therefore the manuscript has some serious flaws. In general, the study is of relatively good quality, rather well organized but needs profound correction to become the interesting and comprehensive reference for the readers. Moreover, the manuscript needs extensive editing of English language and style.

My concerns are as follows:

  • first of all, the antimicrobial profiles of all of the used strains should be given,
  • second of all, the CFEAS bactericidal activity on all of the used strains should be investigated and presented,
  • the information on antimicrobial treatment (either topical or systemic) applied for all the included patients should be added to confirm the exclusive CFEAS activity on the wound healing process,
  • the phrase Gram-positive and -negative microorganisms is wrong. It refers to bacteria only.
  • Line 19, the phrase “Human clinical, microbiological and histological pilot trial” sound unclear for the readers,
  • Line 22, “activity from”?
  • Line 40, the phrase “how and how often” needs correction,
  • Introduction section, at least in my opinion some parts should be rewritten to make it easier to follow for the readers, e.g. lines 59-62,
  • Line 64/65, “covered with platinum” or “platinum covered” would be more suitable,
  • Line 70 and the following, “wound bed” you mean “bedsores”?
  • Line 71, “usefulness”?
  • Line 77-78, the sentence needs correction, it is confusing,
  • Line 78, the definition of “chronic wound” is necessary,
  • Line 81, “bacteria”,
  • M&M, What are the storage conditions of the clinical strains,
  • Line 83, what are the exact quantitative microbiological/diagnostic interpretation criteria for the infected wound?
  • Line 85, the biofilm production methodology should be described, at least briefly,
  • Line 86, MICs of what? This is not methodology for MICs determination. What about the sterility control of this step?
  • Line 91, how did you manage to prepare this particular bacterial concentration (CFU)?
  • Line 92, the exact CFEAS concentrations should be given, not the range,
  • Line 93, the same for PHMB,
  • it should be explained why the exact CFEAS and PHMB concentrations were applied,
  • Line 94, positive control of what?
  • Line 96, and the following, lack of reagents/media manufacturers,
  • Line 98, CFU/gram of what tissue?
  • Line 100, why did you chose two (out of five) experiments for analyses?
  • Line 104, what was the purpose of glucose addition, and this particular glucose concentration?
  • Line 107, positive control of what?
  • Line 116/117, why did you chose two (out of three) experiments for analyses?
  • Chapters 2.3 and 2.4, what about the controls/sterility control of this steps?
  • Line 139, why this particular number of cells was chosen and how did you estimated that?
  • Line 172, “performed twice and expressed as…”?
  • Line 184, knowing that the evaluation of infected wounds is quantitative, the phrase ”processed the same day” is not sufficient. Bacteria could still grow/multiply in this medium under these conditions.
  • Chapters 2.9, what about the sterility control of this step (saline and the whole manual procedure)?
  • Line 197, “macerated”?
  • Line 215, “CFEAS avoids biofilm maturation”, maybe it simply kills bacteria? It was not checked.
  • Figure 1, should be clearly divided into Gram-positive and –negative bacteria to make it easier to follow the results,
  • addition of the timepoints (3h, 6h, 12h, 24 h, etc.) on the corresponding figures/graphs would make them easier to understand,
  • in my opinion, some of the sentences from the "Results" section should be deleted or moved to “Discussion” chapter because for example refer to the results of other authors, e.g. 243, 249, 268-270, 289,
  • Line 280 needs to be rewritten,
  • Line 329, “infected human wound” is more suitable,
  • Line 330, “venous leg”?
  • Line 347-349, this observation should be discussed/explained,
  • Line 407, “hallucal”?
  • italics are missing in a number of places,
  • there are some punctuation errors, typos, lack of dashes, unnecessary dashes in the manuscript that have to be corrected,
  • language style and grammar precision are also poor at some places, making some sentences hard to understand.

Reviewer 2 Report

The manuscript describes the antibacterial activity of controlled flow electrolyzed acid solution (CFEAS) on wound pathogens both in vitro and in a pilot trial. The manuscript is moderately innovative and well-written; however, I reckon that there are several issues that should be resolved before publication.

Major comments

In the Materials and methods section, how many replicates were performed per experiment?

In Figures 1 and 2, please, represent the individual values, the median and interquartile range, and the n of each bacterial species.

Why did the author use lower concentrations than the minimal inhibitory concentration in the cytotoxicity assays? For me, this does not make much sense. There are many antimicrobials that are cytotoxic at antibacterial concentrations. We must consider that this cytotoxicity is transitional, and the potential benefits far outweigh these harms.

Regarding the statement from lines 345 to 346, are the authors sure that this can be concluded from their results obtained?

Line 363-367, how have the authors demonstrated this in the in vitro experiments? This is unclear for me.

I consider that lines 416 and 417 are unnecessary.

Minor comments

Throughout the manuscript, there are many terms that should be in italics, such as in vitro, in vivo, or all the species names of bacteria cited.

Throughout the manuscript, there are molecular formulae whose subindex/superindex are not as such e.g., O2-, H2O2, etc. Please, correct them.

In the Abstract section, please replace “Gram negative” and “Gram positive” with “Gram-negative” and “Gram-positive”.

In line 40, I think that there is one “and” too many.

I would suggest including more epidemiological data (sex, age, type of wounds) related to patients from who the strains were isolated.

In Figures 1 and 2, please, remove the sentence “Differences […] control (p>0.05)”. This has already been said in Materials and Methods.

In Figure 6e, please, correct the units of the y-axe, they must be “CFU/g” and not “CFU”.

Line 412, I think that this compound could be also used in burns.

Reviewer 3 Report

The submitted manuscript investigates antiseptic and biosafety effects of a controlled-flow electrolyzed acid solution as well as some of its molecular mechanisms of action. The topic is highly relevant for the management of infected wounds as the removal of pathogens and microbial biofilms is fundamental. The authors study the subject in sufficient width as both in vitro activity/molecular effects and in vivo efficacy in patients are presented. The manuscript has the necessary basic quality elements for making it suitable for publication. However, there are several major points to be addressed prior to publication.

  1. The Abstract is should be improved as currently it is difficult to read and does not reflect all the relevant elements of the work in clear and comprehensible manner.
  2. The text should be improved from the language point of view since many errors are noted (line 71 word useful should be replaced with usefulness, many word throughout the manuscript are presented with unnecessary hyphens). The sentence in line 368 should be reworded as is very unclear.
  3. The microorganisms used should be identified with some internal strain numbers or markings to ensure that the strain can be identified as a unique strain, can be recognized from other strains and that the same strain is used across the studies.
  4. The dilutions of antiseptics used for MIC determination should be stated and/or how many concentrations were tested. Also, the MIC methodology states that MICs were determined by colony counting while at the same time CLSI protocol is referenced which is based on visual readouts of microbial growth/inhibition in liquid medium. Additionally, the colony counts are expressed as cfu/gram tissue which is not applicable to microdilution methodology described. If actually, biocidal activity was determined than MIC terminology should be avoided and adequate terminology, methodology and details described. The methodology used for different experiments should be clearly specified and described.
  5. The In vitro biofilm elimination methodology should be additionally clarified in the last sentence by stating the conditions used for incubating the antiseptics with the formed biofilms (statically, timings…).
  6. The authors stated that the participants completed Institutional consent form for routine ambulatory procedures but as the regulatory status of the CFEAS (marketed product for wound treatment or not) is not clearly explained, the authors should explain if and why the routine form is considered sufficient. Also, the methodology for CFEAS application should be clarified (gauze wet with CFEAS – what amount, CFEAS concentration (100?)).
  7. The results section requires the most revisions as the presentation of the data has much room for improvement:
    • The MIC results are not adequately presented and no MICs is reported. The reported statement that “…Gram-negative and Gram-positive bacteria were susceptible to the treatment with CFEAS at concentrations higher than 95%” makes little sense since 95% was the highest concentration used. More data should be presented and adequate explanations should be provided as the antibacterial effects represent major data for subsequent analyses and claims for efficacy against planktonic bacteria.
    • Figures are quite difficult to follow as the data is presented in captions while it would be more convenient for the reader to have the legends and graph titles to follow the data presented. Additionally, the graphs that compare different time points would present the data more clearly if the data is presented per organism/cell line. E.g. Figure 2 data, would be more clear if each strain is presented separately with all the timepoints. The same would apply to cell line data. Figures 4 and 5 would benefit from the editing in that respect as well.
    • As per journal instructions "Data not shown" should be avoided and authors are encouraged to publish all observations related to the submitted manuscript as Supplementary Material.
    • The authors do not provide the rationale for using only 1-50% CFEAS in cell viability assay i.e. how this omission affects discussion of biosafety since 95% solutions is used in vivo for patients.
    • Figure 4 (as well as its respective description in methodology section) insufficiently explains the observations, the staining colours as well as the expected changes in colours or indications with arrows (or some other method) to the sections/cells of interest.
    • Figure 6 would benefit from further editing by adding pointers and explanations for the observations: biofilm presentation (where is and how it is visible), histological details (specific cells, bacteria, structures). Average reader is not a histopathologist and cannot recognize these details.
    • Clinical data on patients would be even more valuable if some information on the long term follow-up of patients is presented. Was CFEAS applied only once or repeated dosing was attempted (if, yes for how long), what are the outcomes of the therapy in these patients, were the wounds healed, were they showing the signs of healing?
  8. In the conclusion section, the authors state than only preclinical evaluation should be performed in the future. Based on their results should not also randomized, controlled, prospective trials in humans be performed comparing this method with some standard of care?

Round 2

Reviewer 1 Report

Dear Authors,

the manuscript has been generally nicely corrected. However, some of the changes are mentioned only in the Response to the Reviewer file, not in the newer version. It should be corrected. Moreover:

  1. I need to go back to one issue - How did you manage to prepare this exact bacterial concentration (CFU)? - this issue is still not explained, needs description of how did you calculate it so precisely.
  2. The definition of chronic wound should be included into the manuscript, with an indication what is the minimum time without healing process observation to call the wound chronic.
  3. To the best of my knowledge: "Erythromycin" and "Trimethoprim/sulfamethoxazole" need correction, "P. aeruginosa" is the correct and worldwide used abbreviation and "-positive" and "-negative" should not be written with capital letters (Table 1S).

The rest of the changes made in the paper improved the manuscript significantly.

Reviewer 3 Report

The authors have improved the manuscript and the results are much clearer now. However, there is still room for improvement, as the identification of clinical strains has not been included even though an explanation has been provided. There should be some sort of a label associated with each strain to be able to identify it. If a strain is a part of strain collection, a species identification is not sufficient to identify a particular strain, a numerical label should be stated so that it is possible to trace the identical strain in case an experiment needs to be repeated, a colleague want to check that data or a strain needs to be provided to external lab upon request.

Author Response

This manuscript is a resubmission of an earlier submission. The following is a list of the peer review reports and author responses from that submission.